# CACTUS: Content-Aware Compression and Transmission Using Semantics for Automotive LiDAR Data

**DOI:** 10.3390/s23125611

**Published:** 2023-06-15

**Authors:** Daniele Mari, Elena Camuffo, Simone Milani

**Affiliations:** Department of Information Engineering, University of Padova, Via Gradenigo 6/A, 35131 Padova, Italy; daniele.mari@dei.unipd.it (D.M.); elena.camuffo@dei.unipd.it (E.C.)

**Keywords:** point clouds, LiDAR, compression, transmission, semantic segmentation

## Abstract

Many recent cloud or edge computing strategies for automotive applications require transmitting huge amounts of Light Detection and Ranging (LiDAR) data from terminals to centralized processing units. As a matter of fact, the development of effective Point Cloud (PC) compression strategies that preserve semantic information, which is critical for scene understanding, proves to be crucial. Segmentation and compression have always been treated as two independent tasks; however, since not all the semantic classes are equally important for the end task, this information can be used to guide data transmission. In this paper, we propose Content-Aware Compression and Transmission Using Semantics (CACTUS), which is a coding framework that exploits semantic information to optimize the data transmission, partitioning the original point set into separate data streams. Experimental results show that differently from traditional strategies, the independent coding of semantically consistent point sets preserves class information. Additionally, whenever semantic information needs to be transmitted to the receiver, using the CACTUS strategy leads to gains in terms of compression efficiency, and more in general, it improves the speed and flexibility of the baseline codec used to compress the data.

## 1. Introduction

Sensing technologies have been recently boosted by the availability of versatile 3D acquisition devices that have proved to be capable of generating accurate models of the surrounding scene at a reasonable cost and in a limited time. Within these scenarios, LiDAR technology has played a leading role considering its widespread application in several fields including (but not limited to) land sensing and reconstruction, robotics, cultural heritage, and monitoring [1,2,3]. With respect to automotive applications, LiDAR PCs allow a real-time 3D modeling and semantic understanding of the environment [4] which enable smart applications such as autonomous driving, traffic monitoring, and path planning, to mention some of them.

However, most data processing architectures are unable to elaborate efficiently a large volume of LiDAR data in a reasonable time [5]. In addition, sharing environmental information could be required in order to enable a more efficient scheduling of operation plans, reorganize navigation paths, or access more efficient classification tasks [6,7]. It is possible to overcome this bottleneck by employing some shared highly performing computational resources, moving most of the advanced processing toward cloud computing environments [8,9,10,11].

Moreover, transferring part or most of 3D processing operations from terminals to network processing units implies transmitting big amounts of information, which must preserve its informational and semantic contents in order to be useful for the considered elaboration task (e.g., object classification, semantic segmentation, planning, etc.). Compression proves to be crucial since it allows constraining the impact on computing and transmission facilities (e.g., latency, network congestion, and energy consumption). Indeed, whenever coding operations are too aggressive or unaware of final application, the semantic information of the scene cannot be recovered any more as the data arrive at the processing unit (see results on semantic segmentation reported in Section 4). The compression of terrestrial LiDAR point clouds is a notoriously hard problem in 3D data transmission due to the spatial sparsity and inhomogeneity of acquired 3D points as well as to the noise that affects their coordinates.

For these reasons, aside from the most known PC codecs such as TMC13 [12] and Draco [13], several other approaches have been proposed. As an example, the work by Huang et al. [14] uses an entropy model to exploit the remaining redundancy after an octree decomposition of voxel volume [15]; Tu et al. [16] use Recurrent Neural Networks to progressively encode residuals, while in the work by Varischio et.al. [17], the most meaningful parts of the PC are transmitted after a segmentation of the input PC. Other meaningful approaches that are not specifically targeted toward LiDAR but that are worth mentioning are deep learning (DL)-based Point Cloud Coding (PCC) approaches [18,19,20,21,22,23,24] that generally exploit convolutional autoencoders for efficient features extraction and decoding.

Following these trends, some preliminary works on multimedia compression and transmission have highlighted the fact that cognitive and semantic information can be extremely useful when applied to source coding and multimedia data transmission [25].

The advantage of a semantic-aware coding system is two-fold:Semantic information allows dynamically allocating the bit rate to different parts of the PC and schedules information transmission in a content-aware manner (while discarding useless data).The distortion introduced by coding operations could prevent a correct classification when data are reconstructed and segmented at the receiver (see Section 4); a cognitive source coding approach allows signaling to the receiver semantic information obtained from the originally acquired data.

Most of the works that were explored in the literature address the compression task in a very general way and thus do not exploit semantic data for transmission, since it is not always available. To the best of our knowledge, up to the publication of this paper, the only work that makes use of this additional information is [17]. However, the authors fail to fully exploit it since, in the paper, semantics only serve the purpose of selecting pieces of the PC that should be discarded in order to obtain lower compression rates and faster decoding. As a matter of fact, they do not realize that the pre-computed semantic information can be transmitted together with the geometry almost for free. Additionally, in their case, the semantic information is used to divide the PC very coarsely, but increasing granularity does not really hurt performance and actually provides greater flexibility. For these reasons, we believe that the advantages of using semantic information for guiding compression and transmission still have not been properly explored.

In a nutshell, the current paper proposes a general framework for LiDAR PC compression that first segments the input PC into multiple semantically characterized clusters that are then independently coded and transmitted, tuning the compression parameters according to the significance of the coded class and the desired quality level.

This approach is particularly well suited for LiDAR data, since it is one of the most used sensors in automotive scenarios where the Semantic Segmentation (SS) literature is already well established.

The main novelties and advantages of the proposed approach can be listed as follows:Semantic information can be transmitted with a very limited overhead (see Section 3), sparing the segmentation task at the decoder side.Transmitting labels proves to be more accurate than operating SS at the decoder on the reconstructed PC (because of compression noise).The bitrate can be flexibly allocated by choosing the quality level or skipping some parts in a content-aware manner according to the network conditions or user preferences.The execution time required to compress the PC and the semantic information is lower with CACTUS than with the reference codecs. This result is obtained without considering the time required to compute the semantic information, because this step needs to be computed regardless of the coding procedure.

The framework was implemented using RandLA-Net [26] for the segmentation part, and TMC13 [12], Draco [13], and Distributed Source AutoEncoder (DSAE) [24] for the compression part. Nonetheless, the processing pipeline can be generalized to different SS architectures and codecs. The architectures choice was mainly driven by the fact that the considered algorithm is intended for a mobile automotive scenario, where latency and computational complexity are significantly constrained. In this regard, experimental performances were tested on a subset of SemanticKITTI [27,28], which is one of the most popular benchmarks for autonomous driving.

The paper is organized as follows: in Section 2, the main advancements in collaborative strategies using LiDAR PCs are presented together with an explanation about PCs codecs; then, in Section 3, the proposed method is introduced and analyzed while its performances are examined in Section 4 and all the results are further discussed in Section 5. Finally, in Section 6, the conclusions are drawn, which are followed by some hints for future works. The code for the framework can be found at https://github.com/Dan8991/CACTUS-Content-Aware-Compression-and-Transmission-Using-Semantics-for-Automotive-LiDAR-Data (accessed on 15 January 2023).

## 2. Related Work

Due to the increasing hype that self-driving cars are rising in the industry and academia, a lot has been accomplished to reduce at its minimum the computational and transmission cost of the acquired data. In particular, collaborative strategies have been developed to avoid redundant calculations, while coding algorithms have been designed in order to produce representations that are as efficient and effective as possible, facilitating data exchange among nearby vehicles.

### 2.1. Collaborative Strategies for LiDAR Point Clouds

The idea of sharing LiDAR PC data and processing them in a collaborative and distributed fashion has been recently investigated in many applications ranging from cultural and natural preservation [2,29,30] to automotive applications [31,32].

In [11], Shin et al. propose a road object mapping system for LiDAR data based on an edge–fog–cloud platform. Similarly, it is possible to obtain an HD mapping of the surrounding environment by fusing multiple contributions acquired by diverse vehicles at different locations and instants [33]. The problem of registering and fusing multiple acquisitions (which can have different orientations and distances) can be solved by matching lines and geometrical properties [1] or analyzing semantic information [34,35]. Alongside with reconstruction problems, the approach [10] tackles the problem of localizing the mobile terminal in an indoor environment thanks to an edge-side Simultaneous Localization And Mapping (SLAM) engine. A similar problem is solved in [36] for a swarm of unmanned ground vehicles (UGVs) using a two-stage graph optimization. Similarly, merged LiDAR data are employed for multiple object detection and tracking combining estimation filtering with a discretization of space in [37]. In addition, the solution in [38] adopts some a priori information about the environment and the context to localize cars using some low-complexity estimation strategies.

Together with reconstruction and localization, the semantic analysis of the surrounding environment benefits from collaboration as well. During the last few years, it has become possible to find different collaborative perception and classification frameworks [39,40,41] showing the robustness and the safety level that can be obtained by such applications. Alongside a distributed orchestration of the operations, these collaborative approaches integrate different Deep Learning data processing techniques. The approach in [42] resorts to a reinforcement learning algorithm to create a map of the environment, while a position-channel attention mechanism is adopted in [43] to enhance some LiDAR-related features in order to discard the least-relevant information. A transformer-based architecture is employed in [44] on multispectral LiDAR data for land classification.

Although most of the approaches imply the transmission of PCs to the central processing framework, some solutions resort to federated learning strategies to avoid disseminating the data sensed by the different terminals with all the privacy-related implications [45].

### 2.2. An Overview of Compression Strategies for LiDAR Point Clouds

In the literature, PCC approaches can be divided into two separate classes: traditional source coding solutions [12,46,47] and DL-based codecs [18,19,20,21,22,23,48,49,50]. In the first set of approaches, a compact representation of the geometry is obtained by using an ad hoc data structure (e.g., a graph, a KD-tree [51], an octree [15] or cellular automata [52]) and then entropy coding. Some recent solutions aim at exploiting the temporal redundancy existing between temporally adjacent acquisitions by entailing a bidirectional prediction [53].

In the second set of approaches, instead, deep auto-encoders are usually exploited to generate smaller and entropy-efficient hidden feature representations [18,19,20,21,22,24].

In the work by Huang et al. [14], the PC is quantized and encoded using an octree decomposition where each set of octants can be represented by a byte. Such representation proved to work well on LiDAR PCs since sparsity can be handled efficiently. A deep entropy model is then used to estimate the probability of each symbol, and the resulting values are used as context by an arithmetic coder.

In [16], the PC is represented in a format similar to depth maps. This 2D representation is then compressed with an autoencoder, where a quantizer block maps the latent space features into binary strings. In this scheme, the Neural Network (NN) performs a first compression of the original image; then, some residuals are progressively encoded to refine the accuracy of the reconstructed samples. An LSTM-based solution is proposed in [54] obtaining higher performances with respect to other state-of-the-art PC codecs. However, such approaches are computationally burdensome and do not allow for content-aware compression.

In the work by Varischio et al. [17], Draco [13] is used in conjunction with Rangenet++ [55] to perform the encoding. In particular, the authors define three compression levels where parts of the PC are progressively removed to allow to perform compression in real time. At the highest quality level, the whole PC is coded; at the second one, points related to the street are removed, and at the lowest quality level, only points for critical classes (e.g., person, car, motorcycle) are retained. This codec allows shaping the bitrate according to the network constraints, but the bit allocation may be non-optimal whenever data from all the classes (e.g., road and vegetation) are required. On top of that, semantic labeling is discarded during transmission so the receiver has to recompute it himself on the reconstructed point cloud. The latter has holes and is affected by compression noise; therefore, it is likely to lead to a less accurate semantic segmentation.

## 3. Methodology

The CACTUS pipeline is composed of two main building blocks. The first is designed for a semantic understanding and organization of the acquired samples representing the scene, while the second consists in the core compression strategy that, adequately optimized, is going to create the bit stream. The proposed framework can be represented by the block diagram reported in Figure 1, where the input PC is segmented into separate semantic classes that are independently coded by a PCC architecture. Indeed, in the CACTUS framework, we assume that a SS unit is available on automotive terminals, since it is employed for the navigation or safety issues (e.g., pedestrian detection, obstacle avoidance, et.). As a matter of fact, the acquired point clouds could be semantically segmented and characterized after their acquisition.

In the current implementation, we adopted the architecture RandLA-Net for SS, while standardized codecs TMC13 or Draco were considered for the compression of the acquired point clouds and the formatting of the transmitted bit stream. These segmentation and coding architectures were chosen since they prove to be among the most suitable and standard-compliant solutions for LiDAR point cloud coding, although several other codecs or segmenting networks [56,57] could be adopted as the proposed framework can be generalized very well. Additionally, in order to assess the properties of the framework when using one of the most recent learned codecs, some tests were also carried out with the DSAE [24] network. The data employed belong to the validation sequence of the SemanticKITTI dataset (the first 100 samples of sequence 08).

### 3.1. Adopted Segmentation and Coding Solutions in the CACTUS Framework

**RandLA-Net**: RandLA-Net [26] is probably one of the best-performing architectures for LiDAR SS that is completely point based. In addition, it is one of the most lightweight solutions available, providing good results without needing to voxelize the input data. RandLA-Net tackles the problem that many architectures have in capturing point-wise context information as it proposes an effective local feature aggregation module, which is able to automatically preserve complex local structures by progressively increasing the receptive field for each point. This module can obtain successively larger receptive fields by explicitly considering the local spatial relationship and point features, being more effective and robust for learning complex local patterns. In general, RandLA-Net only relies on random sampling within the network, thereby requiring much less memory and computation. The entire network only consists of shared multi-layer perceptrons, making it ideal for large-scale point clouds.

**TMC13**: TMC13 is an implementation by the MPEG group of the Geometry-based Point Cloud Coding (GPCC) standard; it works by encoding the PC with an octree that is then processed by an arithmetic coder to reduce redundancy. The main way to perform rate control is to quantize the coordinates before entropy coding. In particular, the *codingScale* parameter can be used to control the amount of quantization. In the results section, we refer to a quantization parameter qp where the two are related by the equation $\mathrm{codingScale}=2^{-qp}$. In TMC13, the attributes are encoded using the Region Adaptive Hierarchical Transform (RAHT) transform [58]. In this work, the class is encoded as the color attribute by setting the values $r_{i},g_{i},b_{i}$ relative to point $p_{i}=\left(x_{i},y_{i},z_{i}\right)$ as its semantic class $c_{i}$. This was achieved because, to the best of our knowledge, arbitrary attribute compression is not supported in TMC13 and Draco, but this procedure should be almost equivalent to an ad hoc implementation since the arithmetic coder should be able to completely remove the redundancy due to the repetition of the value over the three channels being deterministic.

**Draco**: Draco is an efficient codec implemented by Google that performs compression based on KD-trees [51] instead of octrees. This data structure has the advantage that it can place the boundaries that split the space where it is most efficient, which allows to better encode point clouds with uneven densities.

Draco with respect to TMC13 has the advantage that it provides the *cl* (compression level) parameter that can be used to reduce the computation time at the price of worse compression ratios. Here, the qp parameter was computed as $qp=15-qp_{d}$, where $qp_{d}$ is the quality parameter defined by Draco. In this case, the class was losslessly encoded as the color likewise TMC13.

**DSAE**: DSAE is a learned geometry codec proposed in [24] based on the principles of distributed source coding [59]. The main idea of this paradigm is that the receiver knows a signal $\hat{x}$, called side information, that is correlated with a signal *x* at the sender side. Because of this, the latter can just transmit some bits *s* called syndromes that the receiver can use to correct his own signal, increasing its similarity with *x*. In this case, the side information is represented by the PC compressed with another codec (TMC1 in the original paper). During transmission, the sender will use a convolutional encoder to compute the syndromes, which will be compressed with an arithmetic coder and transmitted to the receiver. The latter will feed them, together with the side information, to a convolutional decoder that should be able to reconstruct the PC with higher quality with respect to the side information.

For a more detailed description of the approach, we refer to the original paper.

### 3.2. Content-Aware Compression for Automotive LiDAR Point Clouds

This subsection presents the operational pipeline that defines the general backbone of the CACTUS coder. The input point cloud $P=\{p_{1},\ldots ,p_{n}\}$ is processed by the semantic segmentation unit (RandLA-Net), splitting the full acquisition into *k* subsets $P_{i}=\{p_{i,1},\ldots$, $p_{i,n_{i}}\}$, i∈{0,…,k}, where the gathered points belong to the same semantic class $c_{i,j}=i\;\forall \;j$. Empty segments, i.e., such that $|P_{i}|=0$, are skipped. Whenever $|P_{i}|\neq 0$, the point subset is compressed using a configurable PCC generating a binary stream $b_{i}$. Codec parameters for $P_{i}$ can be manually tuned (e.g., the Quantization Parameter (QP)) in order to vary the reconstruction accuracy and the allocated bitrate. This larger amount of degrees of freedom makes the approach more flexible with respect to [17], where three fixed configurations are adopted and only a limited set of information is transmitted. In this way, the user or the algorithm can choose to lower the quality of semantic classes (or even completely remove them) that are less useful in the considered scenario (e.g., a self-driving car might not need to precisely know the position of the vegetation outside the road). This allows the system to be more robust to scenarios when the bandwidth is limited, for example when there is high network congestion.

Moreover, the CACTUS scheme allows transmitting semantic class information alongside with point cloud geometry data, making semantic segmentation unnecessary at the receiver side. The compressed file format can be represented by Figure 2 and consists of a small initial header *h*, which is followed by the sizes ($s_{1},\ldots ,s_{k})$ in bytes of each compressed representation and the corresponding encoded points. Therefore, it is possible to represent the final bit stream with the concatenation $b=concat(\left[h,s_{1},b_{1},s_{2},b_{2},\ldots ,s_{k},b_{k}\right])$. The header *h* is formatted in a 4-byte string, where each bit flags whether the associated semantic class is present in the bit stream or not (in the case of SemanticKITTI, only 20 bits are strictly required). The integer $s_{i}$ is coded by 4 bytes and represents the size of the compressed PC $b_{i}$. The total size of the bitstream can be computed as:(1)$$len\left(b\right)=len\left(h\right)+\sum_{i=1}^{k}\left(len\left(s_{i}\right)+len\left(b_{i}\right)\right)$$
where the len(·) function returns the size in bytes of the argument. Adding the header *h* allows sending semantic information just with an overhead of 4(k+1) bytes, which is almost negligible with respect to the total transmission size. This way, the decoder can obtain all the required information to reconstruct the full PC. In fact, using the file sizes, it can obtain the compressed version of the segments that can be independently decoded by TMC13, Draco, or DSAE without any additional information. Additionally, thanks to the binary flags in the header, it can easily infer the class relative to each PC $P_{i}$.

Note that the proposed architecture is quite general, since any segmentation or compression algorithm can be used depending on computational and accuracy requirements. Experimental results showed that the coding performance is not significantly affected by the choice of the SS algorithm. Indeed, even if some points are misclassified (thus becoming isolated after partitioning the PC), many codecs have some coding options that efficiently handle this eventuality (e.g., direct coding mode in TMC13). On the contrary, compression performance can significantly depend on the codec’s robustness with respect to sparsity and noise. For this reason, experimental data were obtained on TMC13 and Draco when considering standard codecs and with DSAE for learned approaches.

### 3.3. Learned Point Cloud Coding Methods

Before moving to the results section, it is important to briefly explain how DSAE was integrated in CACTUS and to mention the main problems that Learned Point Cloud Geometry Codecs (LPCGCs) approaches currently display when used in CACTUS or more in general when applied to LiDAR PCs.

While Draco and TMC13 are widely tested approaches that are very robust to sparsity and other issues that often affect PCs, the same cannot be said for Deep Learning-based approaches. As a matter of fact, many of these need to work on voxel grids in order to be able to process the data, since this allows for the usage of 3D convolutions. This is, however, not well suited for LiDAR PCs where in the regions close to the sensor the signal is very dense, while when further away, it is very sparse. This means that using a coarse voxel grid would destroy a lot of information close to the center, while using a finer resolution would result in very sparse regions that convolutional neural networks struggle to handle and that would increase the bitrate excessively.

This problem is tackled in TMC13, for example, by using the direct coding mode, but these techniques are not yet implemented in learned codecs, since they are still at very early stages, and researchers are still trying to solve many of the challenges presented by PCs signals.

All the aforementioned problems are slightly amplified by the CACTUS framework, since it further increases the sparsity because of the partitioning induced by SS.

Finally, the most widespread LPCGCs are trained on datasets obtained by sampling the points from a mesh (e.g., [21,22] train on ModelNet40 [60]) or on PCs obtained with stereo systems such as the ones from the MPEG dataset (e.g., [18,23,24]), and very few works focus specifically on LiDAR data. For this reason, most of the LPCGCs suffer from the shift in the input distribution when tested on this scenario.

Ideally, the best way to exploit these codecs with CACTUS would be to train a conditional model that exploits information about the class of the partitioned points to achieve better rate-distortion performance; however, this would require an ad hoc codec which goes against the philosophy of CACTUS that was designed to work without major modifications with most codecs. For this reason, in this work, we use the DSAE codec proposed in [24], with some slight modifications, as the main codec for the system. The main differences with respect to the original paper are two: i.e., TMC13 is used instead of TMC1 to generate the side information (which provides higher quality side information at a lower rate). Second, the syndromes are forced to be binary by applying sigmoid with the addition of uniform noise U(−0.5, 0.5) to force the network to predict either 0 or 1, since this allowed obtaining a similar performance while making it easier for the arithmetic coder to compress the bitstream.

The main advantage that DSAE has with respect to other learned codecs is that it exploits side information encoded with another codec (in this case, TMC13). For this reason, even in very sparse regions, it is still able to reconstruct some points (which other learned codecs fail at doing) and with a very small overhead, since only a few bits of syndromes need to be transmitted.

One final issue with learned point cloud codecs is that they usually do not implement attribute compression. This is due to the fact that it is usually very hard to condition the attributes on the decoded geometry, which often results in unsatisfactory results. Additionally, in this scenario, we would like the compression of the attributes to be lossless. For these reasons, in order to compare the compression performance of CACTUS with that of DSAE, we implement attribute compression by means of RAHT (similarly to what is used in TMC13). All the other details such as the packet structure are the same as above.

## 4. Experimental Results

In both TMC13 and Draco, lossy coding is performed via coordinates quantization, whose intensity is ruled by parameters *codingScale* (quality–rate trade-off) and $qp_{draco}$ (quality), respectively. In order to equalize the testing conditions between the two codecs, a generalized quantization parameter qp is used in the experimental tests. Consequently, $\mathrm{codingScale}=2^{-qp}$ for the TMC13 codec, while $qp=15-qp_{draco}$ for Draco. Since DSAE is applied on top of TMC13, their qp ranges are identical.

The semantic information is treated as a point attribute in the compression implementations (attribute coding). This was far more efficient (in terms of rate) than other coding mechanisms (e.g., arithmetic coding), as they are unaware of the properties of PCs.

### 4.1. Experiments

The results reported in this section have been obtained by averaging compression results on the first 100 point clouds of the SemanticKITTI validation set (sequence 08). Our method is tested on Draco and TMC13, and the coding performance is evaluated through a comparison with the corresponding standard codecs in terms of PSNR D1 (Figure 3) and execution time (Figure 4).

### 4.2. Results on TMC13 and Draco

Instead, the accuracy of segmentation is measured through the mean Intersection over Union (mIoU), which reports how close the obtained segmentation is with respect to some ground-truth data. Since the main problem tacked by the CACTUS framework is the transmission of point cloud geometry while preserving its semantic characterization, some preliminary tests concerning the impact of coding distortion on the segmentation accuracy need to be performed. This analysis is required because most of the SS architectures are trained and tested on clean data (not corrupted by quantization noise). Therefore, in a realistic scenario where compression of the signal is essential, the performance of the SS algorithm on compressed and reconstructed point clouds might be strongly impaired. To the best of our knowledge, compression effects on LiDAR semantic segmentation are objectively analyzed here for the first time.

Before carrying on, it is worth reminding that whenever compression follows semantic segmentation, some points, which were originally distinct, are overlapping because of the quantization of spatial coordinates. Note that in this case, they could have been assigned to different semantic classes (this is likely to happen in regions of transition between different instances), making the computation of the final mIoU very hard. The impact of compression on segmentation is therefore made using only the TMC13 codec, since Draco does not remove overlapping points.

The decimation on the reconstructed point cloud and the quantization noise that affects the spatial coordinates severely impair the accuracy of object classification and localization. This is evaluated by comparing the mIoU between the ground truth and the segmented data before and after the compression. This procedure generates $\mathrm{mIoU}(y_{net}^{qp},y_{gt}^{qp})$ and $\mathrm{mIoU}(y_{codec}^{qp},y_{gt}^{qp})$. Note that:$y_{net}^{qp}$ is the output predicted by the NN by processing the geometry of a PC compressed with factor qp.$y_{codec}^{qp}$ are the labels obtained by assigning to each compressed point the label predicted by RandLA-Net for its nearest neighbor in the original geometry (recoloring in TMC13).$y_{gt}^{qp}$ is the PC recolored using the original ground truth instead of the RandLA-Net prediction.

Figure 5 displays the difference $\Delta \mathrm{mIoU}=\mathrm{mIoU}\left(y_{codec}^{qp},y_{gt}^{qp}\right)-\mathrm{mIoU}\left(y_{net}^{qp},y_{gt}^{qp}\right)$ versus the QP qp. It is possible to notice that even with qp=0, the network has a very big loss in performance mostly due to a rounding of the coordinate values to the chosen unit of measurement (sensor precision). Indeed, even very small quantization noise can affect the segmentation accuracy at the receiver mounting up for higher qp values. This effect was also verified by adding random uniform noise to point coordinates (interval [−50 mm, 50 mm]), leading to a comparable loss and showing that the architecture is very susceptible to noise.

By considering the execution time (omitting the SS step, since it has to be carried out regardless) experimental results (see Figure 4) show that CACTUS is approximately five times faster than TMC13 and two times faster than Draco at encoding and decoding geometry and classes. This is supported by the fact that in general, splitting a problem that can be solved with an algorithm with super-linear complexity (PCC) into independent subproblems (encoding subsets of the PC with the same semantic class) should reduce the overall number of required operations (see Appendix A).

In the case of PCC, the compression algorithms are assumed to have at least O(n) complexity. In fact, they need to process each point at least once, but it appears to be super-linear (e.g., in order to remove duplicate points, TMC13 performs a O(nlogn) sorting operation). So, there are the conditions to assume that splitting the LiDAR signal in *c* classes actually lowers the upper bound to the total number of operations in the coding procedure. This is not guaranteed to reduce the execution time, but it appears to do so with high probability, since empirical results show that on average CACTUS reduces the encoding time. On top of that, the CACTUS workflow is highly parallelizable, as different parts can be independently encoded, leading to a lower execution time when multiple cores are available. This analysis supports the claim that the proposed framework proves to be efficient with respect to state-of-the-art solutions.

The rate-distortion performance of the codec is analyzed using the quality metrics described in [61]. The plots reported in Figure 3 display the Peak Signal to Noise Ratio (PSNR) D1 (measured in dB) of the reconstructed PC versus the size in kB of the coded stream. In our tests, only PSNR D1 is used for quality evaluation, since PSNR D2 implies the estimation of normals which can not be easily performed on sparse point clouds. Indeed, the latter metrics is mainly used for dense point sets.

As expected, coding semantic classes results in higher rates at constant geometry quality. This is due to the addition of attribute information when encoding with TMC13 and Draco in the case of CACTUS, because splitting the PC into subsets makes it harder to fully exploit the spatial redundancy for the encoder. However, whenever semantic information is required by the receiver (or a more precise rate control needs to be entailed), using a CACTUS segmentation-based codec proves to be more efficient, especially in the case of TMC13.

This result holds for all the samples and the Bjontegaard delta rate computed between the curves produced by CACTUS and TMC13 with class attributes are on average $\Delta rate_{i}=-31.16\%$, $\Delta PSNR_{i}=3.52$. Conversely, it does not always hold for Draco where segmentation generally does not yield significant rate reductions. However, on average, CACTUS is superior in execution time, Rate Distortion (RD) trade-off and rate allocation flexibility, reading $\Delta rate_{i}=-5.26\%$, $\Delta PSNR_{i}=0.98$.

Additionally, encoding the whole PC usually results in higher errors in under-represented classes (since a higher percentage of their points is removed because of quantization): for this reason, we also decided to evaluate performance in terms of average per-class PSNR
(2)$$PSNR_{avg}\left(P,P^{comp}\right)=\frac{1}{\left|c(P)\right|}\sum_{i\in c(P)}PSNR\left(P_{i},P_{i}^{comp}\right)$$
where c(P) is the set of classes that can be found in the original point cloud, and $P_{i},P_{i}^{comp}$ are the subsets of points with class $c_{i}$ from *P* and $P^{comp}$, respectively, where the latter is the compressed version of the former. Using this distortion metric, it is possible to obtain the plots in Figure 6a,b (results obtained for TMC13 and Draco are relative to the bitrates obtained without encoding classes). In general, CACTUS achieves better RD performance than its counterparts, even if they have a major advantage in terms of rate, since they do not require encoding classes. This shows that this type of approach is better at preserving the finer details of the less populated classes that do not contribute as much to the total PSNR.

### 4.3. Results with Neural Codecs

As mentioned above, we have tested the CACTUS framework with [24] because thanks to side information, it is more robust to the shift in distribution due to the difference in the statistics of the training data and the LiDAR data. However, most of the results and the considerations that will be drawn later actually apply to most LPCGCs, since they often process PCs in the voxel domain.

In Figure 7, it is possible to see that similarly to the standard codecs case, also that with learned ones CACTUS results in slightly lower RD performance if the receiver is only interested in the geometry. However, as before when semantic information is required, then CACTUS proves to be much more effective than using standard attribute compression algorithms such as RAHT

Even though CACTUS shows similar behaviors in terms of RD performance even when using learned codecs, the same cannot be said when considering the computational cost. As a matter of fact, neural networks are inherently parallelized on specialized hardware, making it harder to assess the actual difference in encoding and decoding time. However, assuming sequential operations, i.e., using a single core on a CPU, CACTUS would result in a higher number of total operations due to the usage of voxelized representation.

As a matter of fact, while splitting the data based on the semantic class does not change the overall number of inputs when using a point-based representation, the same does not hold with voxels, since partitioning the total number of cells in input to the codec can change in different ways. Consider for example a PC with *n* classes; then, the total number of voxels in input to the codec can:Increase: e.g., if the class of each point is assigned at random, then the total number of voxels is going to increase by a factor of *n* after partitioning.Remain constant: if the PC can be perfectly divided in *n* regions with no overlap, then the total number of voxels will not change.Decrease: there are some configurations, although rare, where there are empty regions in the PC that can be ignored after partitioning because of a better estimation of the bounds of the PC, which can lead to a lower number of voxels being fed to the codec.

This makes it much harder to analyze the complexity of the approach when using neural networks. However, in the sequential case, the execution time is likely going to increase with respect to encoding the whole PC in one pass, since perfect partitioning with no region overlap is very unlikely in this scenario. However, when considering parallel computing capabilities, this issue should be considerably mitigated.

## 5. Discussion

From the aforementioned results, it is possible to point out most of the main advantages and the main disadvantages of the proposed approach.
In particular, the main strengths of the CACTUS framework are:
Across all the proposed codecs (see Figure 3a,b and Figure 7), when semantic information needs to be transmitted, it is more effective to use CACTUS than the default codecs. This is likely due to the fact that standard attribute coding procedures are designed for continuous values (such as color and normals), and they are not suited for categorical labels, leading to sub-optimal performance. This is solved by CACTUS, which independently encodes the geometry of different semantic classes, allowing the transmission of this type of information almost for free.From the analysis in Appendix A and the empirical results in Figure 4, it is possible to see that when the points are encoded without changing their representation, and when the algorithm complexity is super-linear (which is almost always the case), then independently compressing partitions of the PC leads to lower encoding and decoding time. This is especially useful in the automotive scenario where working in real time is a very important requirement. This does not always apply for neural codecs, since most of them process voxelized representations. Additionally, since in those cases parallelization is often exploited, the execution time highly depends on the specific PC that is being processed and on the classes distribution.As can be seen from Figure 5, independently transmitting semantically disjoint partitions helps improve the quality of the class information at the receiver side for various reasons. Firstly, encoding all the PC with a standard codec can lead to the removal of duplicate points, which is non-optimal, since important information (such as the presence of a pedestrian) could be removed from the scene, and secondly, the performance of current SS strategies are hindered by compression noise, as shown by the analysis carried out in this paper.This type of approach has much greater flexibility than simply using a standard codec, since it allows developing rate allocation algorithms that can choose different coding parameters for each semantic class depending on the requirements of the application.


There are, however, some drawbacks such as: There are some use cases where CACTUS is sub-optimal. In particular, if all points are equally important for the end task and if the receiver does not care about semantic information, then using the base codec allows achieving better RD performance. This can be easily seen from Figure 3a,b and Figure 7, where the geometry RD curves of TMC13, Draco and DSAE are above the corresponding CACTUS ones. The main reason for this is that the latter further sparsifies the PC, making it hard to fully exploit the spatial correlation in the data.Many LPCGCs struggle when reconstructing sparse PCs, and the issue is further amplified by CACTUS. This makes it non-trivial to seamlessly use them in the framework.

## 6. Conclusions and Future Works

This paper reports a detailed investigation of the problems and the possibilities derived by the combination of semantic segmentation with point cloud coding in a collaborative automotive scenario. Semantic segmentation strategies have proved to be highly susceptible to quantization noise, thus resulting in sub-optimal performance when applied at the receiver side. The current paper also proposes a new coding framework for LiDAR PCs that processes and transmits semantic classes independently, leading to higher flexibility in shaping the bit stream. Simultaneously, the coding time is reduced, reliable semantic data can be provided at the receiver, and a more flexible bitrate allocation is enabled, allowing modulating the reconstruction quality on the semantic class. The transmission of class information proves to be efficient also in rate-distortion terms, since CACTUS RD performance shows a negligible loss in modeling geometry and a compression gain in coding semantic information. Such advantages have been investigated on three different coding schemes; however, the approach is totally generalizable to other codecs or segmentation strategies.

Therefore, if high flexibility is required or some semantic information needs to be transmitted, the proposed framework provides improvements in terms of RD performance, execution time, and mIoU at the receiver. Future research should focus on improving the framework by designing the codec and possibly the SS algorithm to better interact together, and by developing algorithms that are more robust to compression noise, in particular: The codecs that are currently proposed in the literature do not use semantic information when performing compression; therefore, the performance of CACTUS might be considerably improved with an ad hoc codec that is conditioned on the class of the considered partition. One other possibility would be to make the codec auto-regressive, i.e., the decoder could exploit the already decoded parts of the PC to infer some information about the parts that it still has not reconstructed. This usually leads to bitrate savings which could mitigate the fact that independently decoding the partitions makes it harder for the system to exploit spatial redundancy.The effect of other SS schemes on the framework should be explored to asses their effect on the overall coding performance.Furthermore, this work shows that compression noise can considerably hinder the performance of scene understanding algorithms. This should strongly motivate researchers to study some new techniques to make their models more robust to this and other types of noises. For example, training the NNs on compressed data might make them much more robust to the modified data distribution.It might be possible to train the whole system (semantic segmentation model and codec) in an end-to-end fashion to achieve better representational capabilities and faster inference due to the features shared by the two blocks.

## Figures and Tables

**Figure 1 sensors-23-05611-f001:**
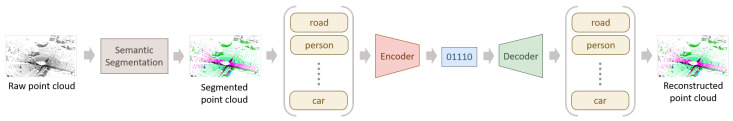
Schematic representation of CACTUS coding process.

**Figure 2 sensors-23-05611-f002:**
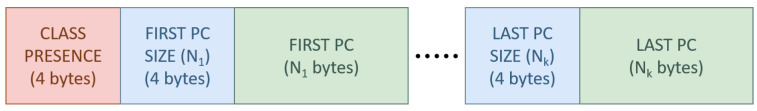
Structure of the PC compressed with the CACTUS.

**Figure 3 sensors-23-05611-f003:**
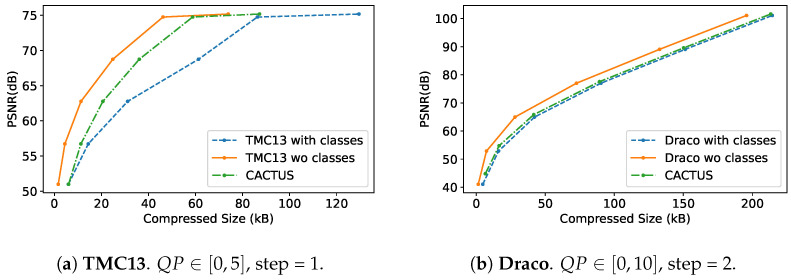
Comparison between TMC13 and Draco rate curves on the full point cloud, with geometry only (orange), with geometry and semantic information (blue), and with CACTUS (green).

**Figure 4 sensors-23-05611-f004:**
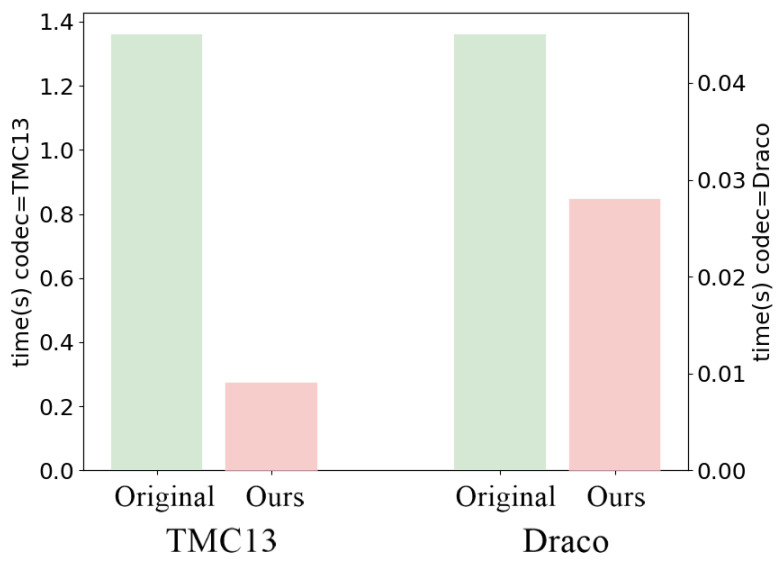
Execution times for CACTUS and the reference codecs.

**Figure 5 sensors-23-05611-f005:**
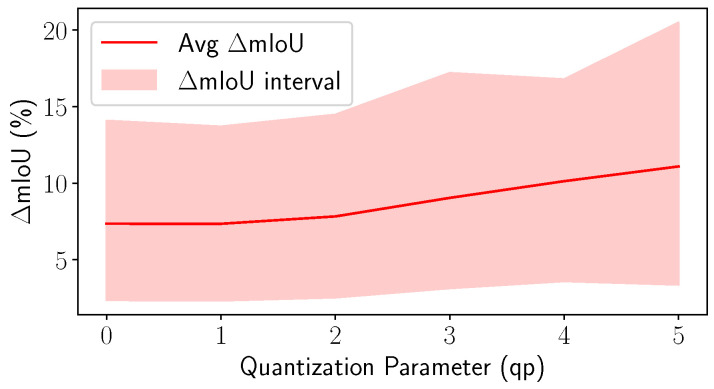
Difference between $\mathrm{mIoU}(y_{codec}^{qp},y_{gt}^{qp})$ and $\mathrm{mIoU}(y_{net}^{qp},y_{gt}^{qp})$.

**Figure 6 sensors-23-05611-f006:**
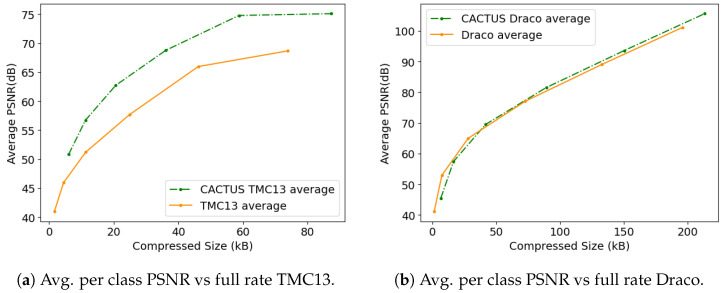
Comparison between the average per class PSNR computed with the standard codecs and with CACTUS.

**Figure 7 sensors-23-05611-f007:**
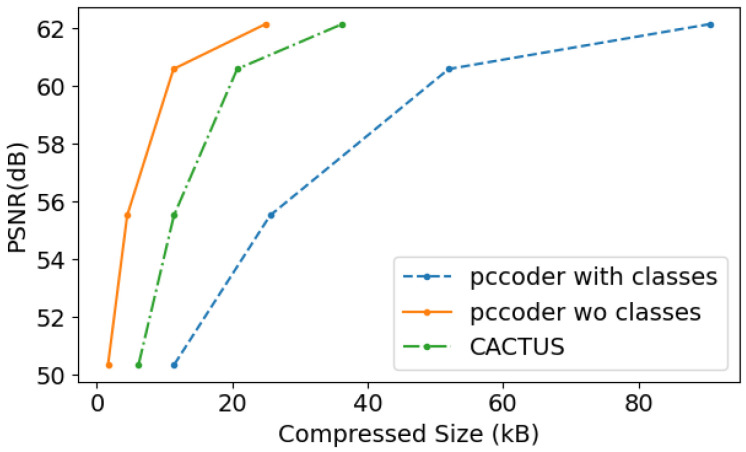
Comparison between rate curves on the full point cloud with geometry only (DSAE), with geometry and semantic information (DSAE + RAHT), and with CACTUS.

## Data Availability

All of the datasets mentioned in this paper are publicly available and properly cited in bibliography. These data can be found here: http://www.semantic-kitti.org/dataset.html accessed date: 15 January 2023.

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
