# Peer review of "CACTUS: Content-Aware Compression and Transmission Using Semantics for Automotive LiDAR Data"

_sensors, 2023, doi:10.3390/s23125611_

Round 1

Reviewer 1 Report (New Reviewer)

The paper proposes a new coding framework that exploits semantic information to optimize the data transmission for Automotive LiDAR data. The subject of the paper is interesting and in line with the aims and scope of the Journal. The paper is well-structured and well-written. However, the paper lacks some important elements such as research gaps, discussion, more detailed future research directions, etc. More detailed comments are provided below.

1.     The abstract should be expanded. It should explain in more detail the background of the problem, as well as the main results, conclusions, and contributions of the paper.

2.     The authors should indicate the research gaps based on the presented literature review.

3.     The paper does not have a proper discussion. The authors did not discuss how the results can be interpreted from the perspective of previous studies. Discussion should clearly and concisely explain the significance of the obtained results to demonstrate the actual contribution of the article to this field of research when compared with the existing and studied literature. In addition, the discussion should point out the limitations, theoretical and practical implications of the study.

4.     Future research directions are weak. There should be at least 3-5 solid future research directions interesting to the majority of the Journal readership.

5.     There are certain technical issues:

a)     There should be at least a couple of sentences between headings of different levels (e.g. between section 2 and sub-section 2.1, etc.).

b)    Acronyms/Abbreviations/Initialisms should be defined the first time they appear in each of three sections: the abstract; the main text; the first figure or table. For example, „LiDAR“ and “CACTUS” are not defined in the abstract. Check the rest of the paper.

English is acceptable.

Author Response

See the attached pdf file.

Reviewer 2 Report (New Reviewer)

The paper introduces CACTUS, a coding framework that utilizes semantic information to optimize the transmission of LiDAR data in automotive applications. By partitioning the point cloud into separate data streams and independently coding semantically consistent point sets, CACTUS preserves class information with limited performance loss and enables flexible bitrate allocation for reconstruction quality. This work is very interesting. Therefore, this manuscript is worth publishing in sensors. However, the author needs to address my concerns as follows.

1.      What are the potential future research activities mentioned in the paper regarding semantic segmentation schemes and countermeasures to compression noise?

2.      How does CACTUS optimize the transmission of LiDAR data?

3.      The English expression may be modified appropriately, and the references should be checked carefully.

4.      What are the advantages of independently coding semantically consistent point sets?

5.      Can the proposed framework be applied to other codecs or segmentation strategies?

The English expression of the article can be modified appropriately.

Author Response

See the attached pdf file

Reviewer 3 Report (New Reviewer)

The authors studied on developing a coding framework exploits semantic information to optimize the data transmission, partitioning the original point set into separate data streams. The current paper proposes a general framework for LiDAR point cloud compression that first segments the input point clouds into multiple semantically-characterized clusters.

The main novelties and advantages of the framework were listed very nice at the end of the introduction part. The number of references are sufficient and they are related to subject of the work. This paper needs some minor revisions before acceptance. I listed some critical issues below for increasing the readability and understandability of this successful study.

1. The abstract section was written well but some key results of the study should be added at the end of the abstract. For example: The key advantages and disadvantages of the proposed framework can be added.

2. Some words in the text were written bold. Why? (For example: Line 168 RandLA-Net or Line 169 TMC13 or Draco etc.) Please check the full text and correct this type of problems.  These words are not bold in other parts of the Text.

3. The authors should check the format of the paper and please use the journal’s paper format. (For example: Line 415- In figure 7 etc.)

4. The caption of Figure 3 needs to check and the authors can embed the description of parts a and b in the caption text.

5. There are abbreviations whose explanation has been forgotten. For example: What does RAHT stand for?

The authors should check all of the text for to fix similar problems.

6. The authors created a abbreviations list at the end of the text. But, there are some missing abbreviations in this list. Please, check this list and complete it with these missing abbreviations such as DSAE, RHAT, LPCGC etc. 

The English level of the paper is good. It needs some minor editting. 

Author Response

Response is reported in the attached pdf file

Round 2

Reviewer 1 Report (New Reviewer)

The authors have invested a substantial effort to address all issues from the previous review round, thus significantly improving the quality of their paper. Therefore, I suggest an acceptance of the paper in its present form.

This manuscript is a resubmission of an earlier submission. The following is a list of the peer review reports and author responses from that submission.

Round 1

Reviewer 1 Report

Very interesting article. Just some minor suggestions:

Place figures next to their references in the manuscript.

line 7

"... **outperforms** the reference codecs, with no **performance** loss."

line 36

"... compression of {terrestrial} LiDAR point clouds is a ..."

line 37-38

Please be careful with the excessive use of "huge amount"

line 64-65

Computational load at decoder side can be disregarded.

line 176

"... without any **discretization** of input data." 

Please note that LiDAR data is a discrete sample of the scene. 

line 292

Separate value and unit with space.

line 352

"... even if the approach ..." please review

Reviewer 2 Report

It is interesting to see such work by combining the semantics with the compression in point cloud processing. However, 

1) MPEG G-PCC and Draco are two traditional codecs. It is suggested to have a learning-based point cloud coder in the study.

2) Considering the semantics, it is better to have a rate vs accuracy in addition to the rate-PSNR.

3) Extensive experiments are required especially for practical applications using real-life data not synthetic data.